# Dydrogesterone as an oral alternative to vaginal progesterone for IVF luteal phase support: A systematic review and individual participant data meta-analysis

Georg Griesinger[1]*, Christophe Blockeel[2], Elke Kahler[3], Claire Pexman-Fieth[4], Jan I. Olofsson[5,6], Stefan Driessen[7], Herman Tournaye[2]

1 Department of Gynecological Endocrinology and Reproductive Medicine, University Hospital of Schleswig-Holstein, Lübeck, Germany, 2 Center for Reproductive Medicine, Universitair Ziekenhuis Brussel, Brussels, Belgium, 3 Established Pharmaceuticals Division, Global Biometrics, Abbott Laboratories GmbH, Hannover, Germany, 4 Established Pharmaceuticals Division, Global Clinical Development, Abbott GmbH, Wiesbaden, Germany, 5 Established Pharmaceuticals Division, Global Medical Affairs, Abbott Products Operations AG, Allschwil, Switzerland, 6 Division of Obstetrics and Gynecology, Department of Women's and Children's Health, Karolinska Institutet, Stockholm, Sweden, 7 Global Biometrics, Established Pharmaceuticals Division, Abbott Healthcare Products BV, Weesp, The Netherlands

* georg.griesinger@uni-luebeck.de

**Data Availability Statement:** All relevant data are within the paper and its Supporting Information files.

## Abstract

The aim of this systematic review and meta-analysis was to conduct a comprehensive assessment of the evidence on the efficacy and safety of oral dydrogesterone versus micronized vaginal progesterone (MVP) for luteal phase support. Embase and MEDLINE were searched for studies that evaluated the effect of luteal phase support with daily administration of oral dydrogesterone (20 to 40 mg) versus MVP capsules (600 to 800 mg) or gel (90 mg) on pregnancy or live birth rates in women undergoing fresh-cycle IVF (protocol registered at PROSPERO [CRD42018105949]). Individual participant data (IPD) were extracted for the primary analysis where available and aggregate data were extracted for the secondary analysis. Nine studies were eligible for inclusion; two studies had suitable IPD (full analysis sample: $n = 1957$). In the meta-analysis of IPD, oral dydrogesterone was associated with a significantly higher chance of ongoing pregnancy at 12 weeks of gestation (odds ratio [OR], 1.32; 95% confidence interval [CI], 1.08 to 1.61; $P = 0.0075$) and live birth (OR, 1.28; 95% CI, 1.04 to 1.57; $P = 0.0214$) compared to MVP. A meta-analysis combining IPD and aggregate data for all nine studies also demonstrated a statistically significant difference between oral dydrogesterone and MVP (pregnancy: OR, 1.16; 95% CI, 1.01 to 1.34; $P = 0.04$; live birth: OR, 1.19; 95% CI, 1.03 to 1.38; $P = 0.02$). Safety parameters were similar between the two groups. Collectively, this study indicates that a higher pregnancy rate and live birth rate may be obtained in women receiving oral dydrogesterone versus MVP for luteal phase support.

**Funding:** The authors declare that the study was supported by Abbott and that one or more of the authors have an affiliation to the company. G.G., H. T. and C.B.'s institution has received investigator fees from Abbott during the conduct of the Lotus I and Lotus II studies. Editorial assistance was provided by Josh Lilly of Alpharmaxim Healthcare Communications, funded by Abbott Established Pharmaceuticals. The funder provided support in the form of salaries for several authors (E.K., C.P.-F., J.I.O., S.D.) but did not have any additional role in the study design, data collection and analysis, decision to publish or preparation of the manuscript. The specific roles of the authors are articulated in the 'author contributions' section.

**Competing interests:** G.G.'s institution has received investigator fees from Abbott during the conduct of the Lotus I and Lotus II studies. Outside of this submitted work, G.G. has received nonfinancial support from MSD, Ferring, Merck Serono, IBSA, Finox, TEVA, Glycotope and Gedeon Richter, as well as personal fees from MSD, Ferring, Merck Serono, IBSA, Finox, TEVA, Glycotope, Vitrolife, NMC Healthcare, ReprodWissen, Biosilu, Gedeon Richter and ZIVA. C.B.'s institution has received investigator fees from Abbott during the conduct of the Lotus I and Lotus II studies. C.B. is the President of the Belgian Society of Reproductive Medicine (unpaid) and Section Editor of Reproductive BioMedicine Online. C.B. has received grants from Ferring, participated in an MSD-sponsored trial and has received consultancy fees from Ferring, MSD, BioMérieux, Abbott and Merck. E.K. is an employee of Abbott Laboratories GmbH, Hannover, Germany and owns shares of Abbott. C.P.-F. is an employee of Abbott GmbH & Co. KG, Wiesbaden, Germany and owns shares in Abbott. J.I.O. is an employee of Abbott Products Operations AG, Allschwil, Switzerland. S. D. is an employee of Abbott Healthcare Products BV, Weesp, The Netherlands and owns shares in Abbott. H.T.'s institution has received investigator fees from Abbott during the conduct of the Lotus I and Lotus II studies. H.T.'s institution has received grants from Merck, MSD, Goodlife, Cook, Roche, CooperSurgical, Besins, Ferring and Allergan. H.T. has received consultancy fees from Gedeon Richter, Merck, Ferring, Abbott and ObsEva. This does not alter our adherence to PLOS ONE policies on sharing data and materials.

## Introduction

Ovarian stimulation regimens using exogenous follicle-stimulating hormone (FSH) preparations, in combination with gonadotrophin-releasing hormone analogues, are integral components of *in vitro* fertilization (IVF) treatment protocols [1]. However, ovarian stimulation can negatively affect the endometrium and the duration of the luteal phase, thereby reducing the possibility of normal implantation and pregnancy [2–4]. In order to improve pregnancy rates obtained from IVF, it is now considered standard practice to support the luteal phase with exogenous progestogens [1, 5, 6].

Progesterone for luteal phase support can be administered via oral, intramuscular, intravaginal [7], subcutaneous [8], or rectal routes [9]. While oral administration is convenient for patients, oral micronized progesterone has relatively low bioavailability and undergoes extensive first-pass metabolism, limiting its use for luteal phase support in IVF [7, 10–12]. To date, progesterone has most commonly been administered via the intramuscular and intravaginal routes [7], and subcutaneous progesterone has recently been introduced into the market [8]. Micronized vaginal progesterone (MVP), administered as capsules or an 8% gel, has been prescribed more frequently than intramuscular progesterone–as shown in a worldwide web survey [7]–as MVP avoids the unpleasant administration-related side effects associated with intramuscular administration (such as injection-site pain and abscess formation) [13, 14]. However, MVP is associated with its own vaginal administration-related side effects, including irritation and discharge [14, 15], and some women may not be comfortable using MVP due to cultural reasons [9]. Recently, MVP has been shown in a pilot IVF study to affect the local microbiome [16]. It has been postulated, following endometrial alterations in the uterine microbiome in response to inflammation, that a progesterone-resistant endometrium can develop [17].

Dydrogesterone is a stereoisomer of progesterone, where the hydrogen atom at carbon 9 is in the β position and the methyl group at carbon 10 is in the α position, the reverse of the progesterone structure (hence denoted "retro" progesterone) [18]. In addition, there is an additional double bond between carbon 6 and 7 whereby the flat steroid configuration is modified, creating a "bent" conformation with enhanced rigidity compared to progesterone [19, 20]. This is thought to account for dydrogesterone having a high selectivity for progesterone receptors with potent progestogenic activity but no or negligible agonistic activity at androgen, glucocorticoid, and mineralocorticoid receptors [21]. In contrast to progesterone, dydrogesterone has higher oral bioavailability [22], which, together with its activity and high specificity for progesterone receptors, along with the efficacy of oral dydrogesterone at a relatively low dose, may minimize side effects [18].

Dydrogesterone, formulated for oral administration, is an alternative to MVP for luteal phase support [23]. Its chemical structure results in high oral bioavailability and increases its specificity for progesterone receptors compared to progesterone [18, 24, 25]. Several studies have indicated that oral dydrogesterone is as efficacious as MVP for luteal phase support [26–32], but these studies were not designed to demonstrate therapeutic equivalence (or non-inferiority). More recently, the large Phase III Lotus I and Lotus II non-inferiority studies, conducted in over 2000 patients in total, demonstrated that oral dydrogesterone was non-inferior to MVP capsules or gel for luteal phase support in fresh-cycle IVF, as determined by pregnancy rates at 12 weeks of gestation [33, 34].

In 2015, a Cochrane systematic review and meta-analysis of aggregate study-level data, comparing oral dydrogesterone (being described as "synthetic" progesterone) versus MVP for luteal phase support in fresh-cycle IVF, found no evidence of differences between groups in live birth rate or ongoing pregnancy rate (two studies, 470 patients) [5]. However, it was

suggested that dydrogesterone was associated with a higher clinical pregnancy rate than micronized progesterone (four studies, 2388 patients) [5]. Following the publication of the Lotus studies [33, 34], it is pertinent to provide a comprehensive summary on the efficacy and safety of oral dydrogesterone versus MVP.

Meta-analyses of individual participant data (IPD), which involve the use of raw patient-level data from each eligible study, have many advantages compared with traditional meta-analyses using aggregate study-level data [35]. For example, meta-analyses of IPD allow for more consistent analyses of data across studies, adjustment of confounding variables, investigation of treatment by covariate interactions, and subgroup analyses [35]. In addition, more precise treatment effect estimates adjusted for significant prognostic factors can be calculated [35]. A one-step approach, whereby the IPD are analyzed simultaneously in a single statistical model that accounts for clustering of subjects within studies, is generally preferred over a two-step approach [36–38].

A one-step meta-analysis of IPD was planned and conducted to synthesize the available efficacy and safety data on oral dydrogesterone versus MVP using a combined dataset from randomized controlled trials (RCTs), and to identify prognostic factors for pregnancy and live birth outcome. Additionally, a straightforward meta-analysis of aggregate data was performed to enable the synthesis of studies where suitable IPD were not available, as well as a meta-analysis whereby the IPD data were combined with aggregate data to improve upon the former analysis.

In summary, the aim of this study was to systematically and comprehensively collate the existing evidence on the efficacy and safety of oral dydrogesterone versus MVP when used as luteal phase support in IVF.

## Materials and methods

### Protocol registration

The protocol for this study is registered with the international prospective register of systematic reviews (PROSPERO; CRD42018105949). This study followed the recommendations of the Preferred Reporting Items for Systematic Reviews and Meta-Analyses (PRISMA) statement [39].

### Eligibility criteria

The inclusion criteria were: 1) prospective RCTs (double-blind, single-blind, or open-label); 2) studies including women undergoing IVF with fresh embryo transfer; and 3) studies comparing the efficacy of oral dydrogesterone (20 to 40 mg daily) with MVP capsules (600 to 800 mg daily) or 8% MVP gel (90 mg daily) by evaluating pregnancy rates (ongoing pregnancy rates for the main meta-analysis of IPD; both ongoing and clinical pregnancy rates for the secondary meta-analysis of aggregate data) or live birth rates. Only studies with available IPD and informed consent from patients allowing the sharing of data with other investigators were included in the meta-analysis of IPD. All eligible studies, including those without suitable IPD, were included in the meta-analysis of aggregate study-level data.

Studies including women undergoing IVF with frozen-thawed embryo transfer were not eligible for inclusion. Review articles, animal studies, retrospective studies, observational studies, non-randomized studies, and conference abstracts were excluded.

### Information sources and search details

A comprehensive literature search was completed using Embase and MEDLINE databases on the Dialog platform for relevant articles published before 31 March 2020 that met the inclusion

criteria. The following search details were used for Embase and MEDLINE: ((dydrogesterone or duphaston or dabroston or dufaston or terolut or isopregnenone or dehydrogesterone) and (mesh.exact("progesterone") or emb.exact("progesterone") or (progesterone))) and (luteal) and (mesh.exact("in vitro fertilization") or emb.exact("in vitro fertilization") or ("in vitro fertilization" or "in-vitro fertilization" or "test-tube fertilization" or "test tube fertilization" or IVF or ICSI or embryo or blastocyst or oocyte or egg)) not ((frozen or "meta-analysis") and dtype (clinical trial)).

## Study selection

The initial results from the literature searches were screened against the pre-established inclusion criteria through evaluation of titles and abstracts by two or more review team members. Excluded articles included studies that did not report the prespecified outcome measures of interest, review articles, retrospective studies, case reports, and conference abstracts. The full texts of the remaining articles were examined in detail to determine their suitability for inclusion. All investigators of studies that met the inclusion criteria (see Eligibility Criteria section) were contacted to request access to IPD.

## Data collection

Country, study design, sample sizes, intervention, available parameters, ongoing pregnancy, and live birth rates were collected for all eligible studies. For studies with available IPD, data from the full analysis sample (FAS) and safety sample were collected, and each sample subset was combined. The primary efficacy outcome in the meta-analysis of IPD was ongoing pregnancy rate (FAS). Secondary IPD outcomes included: live birth rate (FAS), incidence of maternal adverse events (AEs) $\geq$ 2% (safety sample), incidence of AEs $\geq$ 2% diagnosed in utero or after delivery in newborns, and incidence of AEs associated with congenital, familial, and genetic disorders (FAS). For aggregate data, absolute treatment differences and odds ratios (ORs) for pregnancy rates and live birth rates were analyzed.

## Analysis and summary measures

**IPD.** In order to identify potential prognostic factors in the respective studies, each of the following factors were compared individually between patients achieving or not achieving an ongoing pregnancy at 12 weeks of gestation or a live birth (derived from simple logistic regression with the respective variable as a single factor): age, country, study site, body mass index (BMI; categorized $< 24$ kg/m$^2$, $\geq 24$ to $< 28$ kg/m$^2$, $\geq 28$ kg/m$^2$), number of embryos transferred, day of embryo transfer, treatment group, and use or not of intracytoplasmic sperm injection (ICSI) (S1 Table in S1 File). The following rule was applied for the logistic regression model with a stepwise selection: if a comparison resulted in a cutoff probability of $\leq 0.3$, then the prognostic factor was included in the model; the cutoff probability for removing factors from the model was set at 0.35. Study site (not country) was chosen to be included in the model as these were highly correlated (collinearity). The logistic regression analysis was performed separately for ongoing pregnancy rate and live birth rate as dependent outcome variables. Additionally, interactions of treatment group with the factors age, site, and day of embryo transfer were added to the model selection procedure to determine differential treatment effects dependent on the value of the factor.

For variables selected as significant prognostic factors of ongoing pregnancy or live birth from the stepwise selection procedure, ORs, 95% confidence intervals (CIs), and *P*-values were calculated from the stepwise logistic regression model, which included the significant factors as well as non-significant factors (with a *P*-value $< 0.35$). For variables not selected as

significant prognostic factors, ORs, 95% CIs, and *P*-values were calculated in a separate regression model, including the significant factors as well as the respective non-significant factors. Statistical significance was defined as a *P*-value < 0.05. For safety parameters, only pooled results using descriptive statistics are presented.

**Aggregate data.** Absolute differences in pregnancy rate or live birth rate between the oral dydrogesterone and MVP groups were calculated for each study, together with 95% CIs. The overall risk difference (RD; 95% CI) between treatment groups and OR (95% CI) were calculated using the inverse-variance fixed and random effects approach. For comparison, meta-analyses of aggregate data were performed on studies that provided suitable IPD as well as all studies that were initially eligible for inclusion in the review.

**Combining IPD and aggregate data.** A two-stage approach was applied. First, the IPD (where available) were reduced to aggregate data (OR and 95% CI) in each study, separately, using stepwise logistic regression as described in the IPD section above. In the next stage, the aggregate data converted from the IPD were combined with aggregate data from the remaining studies, that did not have IPD available, using the inverse-variance fixed and random effects approach [40].

## Risk of bias

Risk of bias across studies was minimized by performing a comprehensive search for eligible studies and by preventing the duplication of data. Risk of bias within studies was determined by assessing the following domains as determined by the Cochrane tool for assessing risk of bias in RCTs [41]: random sequence generation, allocation concealment, blinding of participants and personnel, blinding of outcome assessment, handling of incomplete outcome data, selective reporting, and any other bias. Risk of bias assessments were performed by two or more review team members and differences were resolved by discussion.

## Results

### Study selection

The study selection process is summarized in Fig 1. In total, 84 references were retrieved from the database searches and were screened against the eligibility criteria. Nine studies were initially determined to be eligible for inclusion; of these, suitable IPD were available for two studies [33, 34] (FAS: *n* = 1957; safety sample: *n* = 2059). For seven studies, no IPD or unusable IPD were available after contact with the investigators.

### Study characteristics and risk of bias

The two studies included in the meta-analysis of IPD were Phase III, randomized, multicenter clinical studies evaluating the efficacy and safety of oral dydrogesterone 30 mg (10 mg three times daily [TID]) versus those of MVP capsules 600 mg (200 mg TID) (Lotus I; NCT01850030) [34] or 8% MVP gel 90 mg once daily (Lotus II; NCT02491437) [33] for luteal phase support in fresh-cycle IVF; both studies involved authors of this meta-analysis as investigators. Lotus I was a double-blind, double-dummy study [34], whereas Lotus II was an open-label study, as it was not feasible to make a placebo applicator for the 8% MVP gel [33]. A total of 2065 premenopausal women (> 18 to < 42 years of age), with a documented history of infertility and who were planning to undergo IVF with or without ICSI, were enrolled in the studies [33, 34]. The day of embryo transfer in both studies was determined by the investigator and based on routine practice at the respective clinic. In both studies, the primary outcome was ongoing pregnancy rate at 12 weeks of gestation, as determined by the presence of a fetal

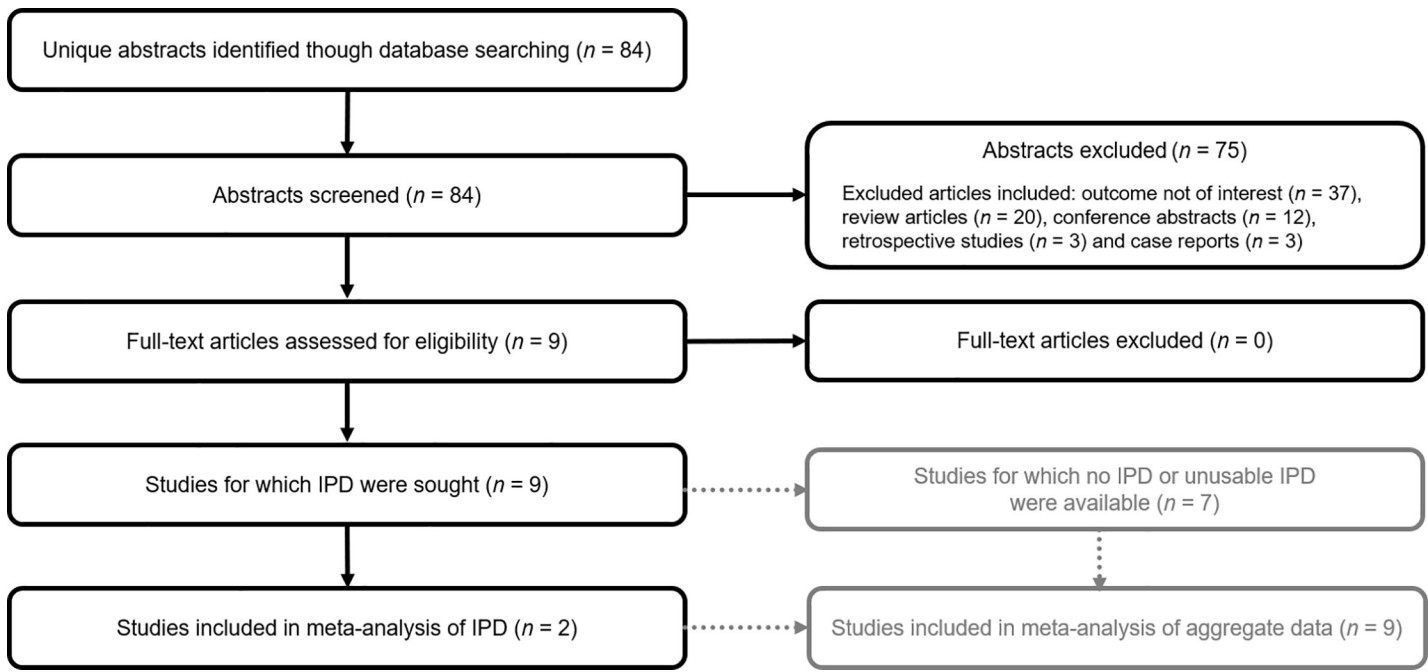

**Fig 1. Flow diagram of the process of selecting and excluding studies for the meta-analyses.** IPD, individual participant data; IVF, *in vitro* fertilization.

heartbeat using a transvaginal ultrasound; a key secondary outcome was live birth rate [33, 34]. The safety outcomes included evaluation of fetal/neonatal and maternal AEs during the study period [33, 34]. Characteristics of all nine eligible studies are shown in Table 1, and S2 Table in S1 File provides an overall summary of the IPD, aggregate data, and combined IPD and aggregate data meta-analyses conducted in this study. There were several different doses of oral dydrogesterone used in the studies. Four studies prescribed 30 mg/day [27, 28, 33, 34]; three studies prescribed 20 mg/day for luteal phase support, and this was in line with the registered dosing for "treatment of infertility due to luteal insufficiency" [26, 29, 30]. The two studies that prescribed 40 mg/day also prescribed a higher dose of MVP [31, 32]. We do not think that the dosage differences influence the findings of this analysis, with its main conclusion on route of administration and drug type, and not dosage. In five of the nine eligible studies, luteal phase support was commenced on the day of oocyte retrieval and continued until 12 weeks of gestation [29, 31–34]. In the study by *Patki et al.* [28], luteal phase support was also started on the day of oocyte retrieval; however, the duration of treatment was unclear. In two studies, it was started on the day of embryo transfer and continued until 12 weeks of gestation [26, 30]. In the study by *Zargar et al.* [27], there was no documentation regarding the day luteal phase support was commenced, but treatment was continued until 12 weeks of gestation.

A summary of the risk of bias [41] for all nine studies is provided in Fig 2. Lotus I demonstrated a low risk of bias across all domains [34], whereas Lotus II scored a high risk of bias in two domains (blinding of participants and personnel, and blinding of outcome data) due to its open-label study design [33]. For the other seven studies [26–32], a high risk of bias was identified in at least three domains (blinding of participants and personnel, blinding of outcome data, and incomplete data). Additionally, of these seven studies, four showed unclear random sequence generation [26, 28, 31, 32], three showed unclear allocation concealment [26, 28, 31], one scored a high risk of bias in the selective reporting domain [30], one showed unclear selective reporting [27], and one scored a high risk of bias in the other bias domain [26]. All studies,

**Table 1. Characteristics of the eligible studies.**

| Study | Country | Study design | Study size | Intervention | Control | Length of intervention | Outcomes | Availability of suitable IPD |
|---|---|---|---|---|---|---|---|---|
| Chakravarty et al. (2005) [26] | India | Randomized, parallel group study | N = 430 | Oral dydrogesterone BID 20 mg/day (n = 79) | MVP capsules TID 600 mg/day (n = 351) | Day of ET until 12 weeks of gestation | Clinical pregnancy Live birth | No |
| Patki et al. (2007) [28] | India | Randomized, parallel group study | N = 675 | Oral dydrogesterone TID 30 mg/day (n = 366) | MVP capsules TID 600 mg/day (n = 309) | From day of OR[a] | Clinical pregnancy Live birth | No |
| Ganesh et al. (2011) [30] | India | Randomized, single-blind, parallel group study | N = 1363 | Oral dydrogesterone BID 20 mg/day (n = 422) | MVP capsules TID 600 mg/day (n = 459); 8% MVP gel OD 90 mg/day (n = 482) | Day of ET until 12 weeks of gestation | Clinical pregnancy | No |
| Salehpour et al. (2013) [32] | Iran | Randomized, single-blind, parallel group study | N = 80 | Oral dydrogesterone QID 40 mg/day (n = 40) | MVP capsules BID 800 mg/day (n = 40) | Day of OR until 12 weeks of gestation | Clinical pregnancy | No |
| Tomic et al. (2015) [29] | Croatia | Randomized, double-blind[b] parallel-group study | N = 853 | Oral dydrogesterone BID 20 mg/day (n = 426) | 8% MVP gel OD 90 mg/day (n = 427) | Day of OR for 10 weeks | Ongoing pregnancy | No |
| Saharkhiz et al. (2016) [31] | Iran | Open-label, randomized, parallel-group study | N = 234 | Oral dydrogesterone BID 40 mg/day (n = 117) | MVP capsules BID 800 mg/day (n = 117) | Day of OR until 12 weeks of gestation | Clinical pregnancy | No |
| Zargar et al. (2016) [27] | Iran | Randomized, double-blind[b] parallel-group study | N = 612[c] | Oral dydrogesterone TID 30 mg/day (n = 212) | MVP capsules BID 800 mg/day (n = 200) | Until 12 weeks of gestation[d] | Ongoing pregnancy | No |
| Tournaye et al. (2017) [34] | Seven countries[e] | Multicenter, randomized, double-blind, double-dummy, parallel-group study | N = 1031 | Oral dydrogesterone TID 30 mg/day (n = 520) | MVP capsules TID 600 mg/day (n = 511) | Day of OR for 10 weeks | Ongoing pregnancy Live birth | Yes |
| Griesinger et al. (2018) [33] | Ten countries[f] | Multicenter, randomized, open-label, parallel group study | N = 1034 | Oral dydrogesterone TID 30 mg/day (n = 520) | 8% MVP gel OD 90 mg/day (n = 514) | Day of OR for 10 weeks | Ongoing pregnancy Live birth | Yes |

BID, twice daily; ET, embryo transfer; IPD, individual participant data; MVP, micronized vaginal progesterone; OD, once daily; OR, oocyte retrieval; TID, three times daily; QID, four times daily.

[a]Length of interventions was unclear.

[b]Patients were aware of the treatment arm due to the different routes of administration and the lack of use of a placebo dummy.

[c]Study included an intramuscular progesterone arm (n = 200).

[d]Timing for the start of interventions was unclear.

[e]Austria, Belgium, Germany, Finland, Israel, Russia, and Spain.

[f]Australia, Belgium, China, Germany, Hong Kong, India, Russia, Singapore, Thailand, and Ukraine.

except Lotus I and II, showed a high or unclear risk of bias in the incomplete outcome data domain with regard to reporting of AEs.

## Meta-analysis of IPD

**Demographics and treatment characteristics.** Subject demographics were comparable between the two treatment groups within the combined IPD of the two included studies (Table 2). A high proportion of subjects were < 35 years of age (67.0%) and Caucasian (72.5%), and the mean BMI of the subjects was 23.1 kg/m$^2$. Treatment characteristics were also similar between the two treatment groups (Table 2). A high proportion of subjects underwent ICSI (68.1%), embryo transfer < 5 days (cleavage stage) after oocyte retrieval (65.7%), and had two embryos transferred (60.3%).

**Ongoing pregnancy rate.** In the studies with available IPD, 38.1% (378/991) and 34.1% (329/966) of subjects in the oral dydrogesterone and MVP groups, respectively, achieved an

**Risk of bias (domains)**

| Study | A | B | C | D | E | F | G |
|---|---|---|---|---|---|---|---|
| Chakravarty et al., 2005 [26] | Unclear | Unclear | High | High[a] | High[b] | Low | High[c] |
| Patki et al., 2007 [28] | Unclear | Unclear | High | High[a] | High[b] | Low | Low |
| Ganesh et al., 2011 [30] | Low | Low | High[d] | High[a] | High[b] | High | Low |
| Salehpour et al., 2013 [32] | Unclear | Low | High[d] | High[a] | High[b] | Low | Low |
| Tomic et al., 2015 [29] | Low | Low | High[e] | High[a] | High[b] | Low | Low |
| Saharkhiz et al., 2016 [31] | Unclear | Unclear | High[f] | High[a] | High[b,g] | Low | Low |
| Zargar et al., 2016 [27] | Low | Low | High[e] | High[a] | High | Unclear | Low |
| Tournaye et al., 2017 [34] | Low | Low | Low[h] | Low | Low | Low | Low |
| Griesinger et al., 2018 [33] | Low | Low | High[f] | High | Low | Low | Low |

**Fig 2. Risk of bias for the eligible studies.** Risk of bias legend: A = random sequence generation; B = allocation concealment; C = blinding of participants and personnel; D = blinding of outcome data; E = incomplete data; F = selective reporting; G = other bias. [a]high risk of bias was expected for the assessment of adverse events; the risk of bias was lower for efficacy outcomes due to the objective methods of assessment. [b]A high risk of bias was expected for the reporting of adverse events. [c]Study contained a larger proportion of women > 40 years of age in the oral dydrogesterone group. [d]Single-blind study. [e]Double-blind study, but patients were aware of the treatment arm due to the different routes of administration and the lack of use of a placebo dummy. [f]Open-label study. [g]10.3% participants were excluded after randomization, and the numbers lost to follow-up were unbalanced between treatment groups. [h]Double-blind, double-dummy study.

ongoing pregnancy at 12 weeks of gestation (Table 3). The meta-analysis of IPD demonstrated that subjects receiving oral dydrogesterone had significantly greater odds of ongoing pregnancy compared with those that received MVP (Fig 3; Table 3; OR, 1.32; 95% CI, 1.08 to 1.61; $P = 0.0075$).

Other than the drug treatment (progestogen) arm, the meta-analysis of IPD also identified that maternal age ($P < 0.0001$), study site ($P < 0.0001$), and day of embryo transfer ($P = 0.0003$) were significant prognostic factors for ongoing pregnancy (Fig 3; Table 3). Older subjects had lower odds of ongoing pregnancy compared with younger subjects (OR, 0.95; 95% CI, 0.93 to 0.98). Embryo transfers that took place $\geq 5$ days after oocyte retrieval (blastocyst stage) were associated with higher odds of ongoing pregnancy compared with those that took place $< 5$ days after oocyte retrieval (cleavage stage) (OR, 1.25; 95% CI, 1.11 to 1.41). BMI was not identified as a significant prognostic factor of ongoing pregnancy (Table 3), and the analysis excluded the number of embryos transferred and use of ICSI in the final model as the significance level was greater than 0.35. No significant interactions were found between treatment group and age, site, or day of embryo transfer ($P > 0.10$ for all interactions), strengthening the validity of the model and the consistency of treatment effects across levels of the other factors.

**Live birth rate.** In the oral dydrogesterone and MVP groups, 34.5% (342/991) and 31.2% (301/966) of subjects, respectively, achieved a live birth in the studies with available IPD (S3

**Table 2. Meta-analysis of IPD: Overall demographics and course and outcomes of pregnancy of subjects in the two studies (FAS).**

| Category | Oral DYD (n = 991) | MVP (n = 966) | Total (N = 1957) |
|---|---|---|---|
| Mean age, years (SD) | 32.2 (4.5) | 32.1 (4.5) | 32.1 (4.5) |
| Age category, n (%) | | | |
| < 35 years of age | 664 (67.0) | 647 (67.0) | 1311 (67.0) |
| ≥ 35 years of age | 327 (33.0) | 319 (33.0) | 646 (33.0) |
| Race or ethnicity, n (%) | | | |
| Caucasian | 721 (72.8) | 699 (72.4) | 1420 (72.6) |
| Asian | 253 (25.5) | 245 (25.4) | 498 (25.4) |
| Other | 17 (1.7) | 22 (2.3) | 39 (2.0) |
| Mean BMI, kg/m² (SD) | 23.2 (3.1) | 23.1 (3.0) | 23.1 (3.1) |
| Subjects who underwent embryo transfer, n | 988[a] | 966 | 1954 |
| Subjects who underwent embryo transfer after ICSI, n (%)[b] | 689 (69.7) | 642 (66.5) | 1331 (68.1) |
| Day of embryo transfer after oocyte retrieval, n (%)[b] | | | |
| < 5 days (cleavage stage) | 669 (67.7) | 614 (63.6) | 1283 (65.7) |
| ≥ 5 days (blastocyst stage) | 319 (32.3) | 352 (36.4) | 671 (34.3) |
| Number of embryos transferred, n (%)[b] | | | |
| 1 | 374 (37.9) | 381 (39.4) | 755 (38.6) |
| 2 | 602 (60.9) | 576 (59.6) | 1178 (60.3) |
| > 2[c] | 12 (1.2) | 9 (0.9) | 21 (1.1) |
| Subjects who had at least one newborn, n | 342 | 302 | 644 |
| One newborn infant, n (%)[d] | 267 (78.1) | 257 (85.1) | 524 (81.4) |
| Two newborn infants, n (%)[d] | 74 (21.6) | 44 (14.6) | 118 (18.3) |
| More than two newborn infants, n (%)[d] | 1 (0.3) | 1 (0.3) | 2 (0.3) |
| Subjects who delivered at term (≥37 weeks of gestation), n (%)[d] | 266 (77.8) | 247 (81.8) | 513 (79.7) |
| Subjects who delivered preterm (>22 and <37 weeks of gestation), n (%)[d] | 76 (22.2) | 55 (18.2) | 131 (20.3) |
| Singleton preterm deliveries, n (%)[e] | 34 (44.7) | 25 (45.5) | 59 (45.0) |
| Multiple preterm deliveries, n (%)[e] | 42 (55.3) | 30 (54.5) | 72 (55.0) |
| Total number of newborns, n | 418 | 348 | 766 |

BMI, body mass index; DYD, dydrogesterone; FAS, full analysis sample; ICSI, intracytoplasmic sperm injection; MVP, micronized vaginal progesterone; SD, standard deviation.

[a]Three subjects in the oral dydrogesterone group from Lotus II were discontinued prior to embryo transfer due to study drug-related issues; these subjects were included in the FAS as failures (not pregnant).

[b]Percentages were calculated according to the number of subjects in the FAS who received an embryo transfer in the respective oral dydrogesterone and MVP groups.

[c]More than two embryo transfers were handled as a protocol deviation in both studies.

[d]Percentages calculated according to the number of subjects who had at least one newborn at delivery.

[e]Percentages calculated according to the number of subjects who had at least one preterm delivery.

Table in S1 File). The meta-analysis of IPD identified that subjects receiving oral dydrogesterone had significantly greater odds of a live birth compared with those that received MVP (Fig 3; S3 Table in S1 File; OR, 1.28; 95% CI, 1.04 to 1.57; P = 0.0214).

Other than the drug treatment (progestogen) arm, maternal age (P = 0.0032), study site (P < 0.0001), and day of embryo transfer (P = 0.0002) were identified as significant prognostic factors of live birth (Fig 3; S3 Table in S1 File). Older subjects had lower odds of live birth compared with younger subjects (OR, 0.96; 95% CI, 0.94 to 0.99). Embryo transfers that took place ≥ 5 days after oocyte retrieval (blastocyst stage) were associated with higher odds of live birth compared with those that took place < 5 days after oocyte retrieval (cleavage stage) (OR, 1.27; 95% CI, 1.12 to 1.44). The number of embryos transferred was not identified as a significant prognostic factor of live birth (S3 Table in S1 File), and the analysis excluded BMI and the

**Table 3. Meta-analysis of IPD: Influence of predictor variables on ongoing pregnancy rate (FAS)[a].**

| Variable | Parameter | | Pregnant[b] | | OR (95% CI) | P-value |
|---|---|---|---|---|---|---|
| | | | Yes | No | | |
| *Significant predictor variables*[c] | | | | | | |
| Treatment | Oral DYD | n/N (%) | 378/991 (38.1) | 613/991 (61.9) | Oral DYD vs MVP: 1.32 (1.08 to 1.61) | P = 0.0075 |
| | MVP | n/N (%) | 329/966 (34.1) | 637/966 (65.9) | | |
| Age, years | N Mean (SD) | | 707 31.5 (4.3) | 1250 32.5 (4.6) | 0.95 (0.93 to 0.98) | P < 0.0001 |
| Study site | | | NA[d] | | | P < 0.0001 |
| Day of embryo transfer | < Day 5 | n/N (%) | 437/707 (61.8) | 849/1250 (67.9) | ≥ Day 5 vs < Day 5: 1.25 (1.11 to 1.41) | P = 0.0003 |
| | ≥ Day 5 | n/N (%) | 270/707 (38.2) | 401/1250 (32.1) | | |
| *Non-significant variable* | | | | | | |
| BMI, kg/m² | N Mean (SD) | | 707 23.1 (2.9) | 1248 23.2 (3.2) | BMI < 24 vs ≥ 28: 1.05 (0.72 to 1.52) | P = 0.0820 |
| | | | | | BMI ≥ 24 and < 28 vs ≥ 28: 1.35 (0.91 to 2.01) | |

BMI, body mass index; CI, confidence interval; DYD, dydrogesterone; FAS, full analysis sample; IPD, individual participant data; MVP, micronized vaginal progesterone; NA, not applicable; OR, odds ratio; SD, standard deviation.

[a]Three subjects in the oral dydrogesterone group were discontinued prior to embryo transfer due to study drug-related issues; these subjects were included in the FAS as failures (not pregnant).

[b]At 12 weeks of gestation.

[c]ORs, 95% CIs, and P-values were calculated by logistic regression analysis for all variables included in the final model of the stepwise selection procedure.

[d]75 sites in the dataset.

use of ICSI from the final model as the significance level was greater than 0.35. No significant interactions were found between treatment group and age, site, or day of embryo transfer (P > 0.10 for all interactions).

## Meta-analysis of combined IPD and aggregate data

For the two studies with available IPD, stepwise logistic regression was applied to calculate ORs and 95% CIs as an aggregate treatment effect. These ORs were then combined with aggregate data from the remaining seven studies. This analysis showed that the OR was statistically significant, both in the fixed effect and random effects models, for both pregnancy and live birth rates (Fig 4; pregnancy rate OR, 1.16; 95% CI, 1.01 to 1.34; P = 0.04; live birth rate OR, 1.19; 95% CI, 1.03 to 1.38; P = 0.02; random effects model).

## Meta-analysis of aggregate data

For the two studies with available IPD, a meta-analysis of aggregate data was also performed. This analysis identified a numerically higher RD and odds for ongoing pregnancy (S1 Fig in S1 File; RD, 0.04 [4%], 95% CI, 0.00 to 0.08; P = 0.06; OR, 1.19; 95% CI, 0.99 to 1.44; P = 0.06) and live birth (S2 Fig in S1 File; RD, 0.03 [3%], 95% CI, –0.01 to 0.08; P = 0.11; OR, 1.16; 95% CI, 0.96 to 1.41; P = 0.11) with oral dydrogesterone versus MVP, although these were not statistically significant.

The meta-analysis of aggregate data was extended to include all eligible studies (nine reported pregnancy rates and five reported live birth rates). While there was a trend for higher pregnancy rates with oral dydrogesterone versus MVP, the RD and OR were not statistically significant (S3 Fig in S1 File; RD, 0.03 [3%]; 95% CI, 0.00 to 0.05; P = 0.08; OR, 1.13; 95% CI, 1.00 to 1.28; P = 0.06; random effects model). Similar results were obtained for live birth rates (S4 Fig in S1 File; RD, 0.03 [3%]; 95% CI, 0.00 to 0.06; P = 0.07; OR, 1.14; 95% CI, 0.99 to 1.32; P = 0.06; random effects model).

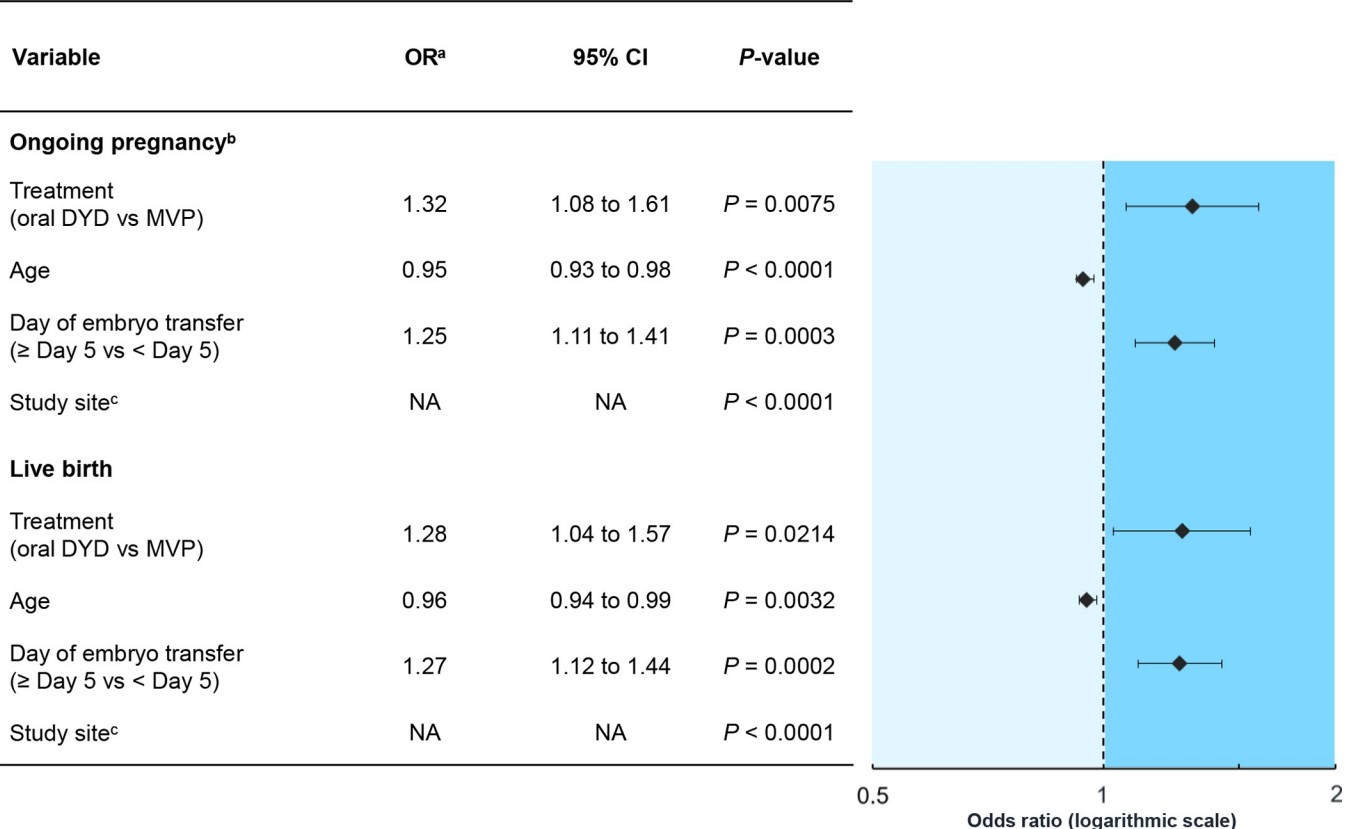

| Variable | OR[a] | 95% CI | P-value |
|---|---|---|---|
| **Ongoing pregnancy[b]** | | | |
| Treatment (oral DYD vs MVP) | 1.32 | 1.08 to 1.61 | P = 0.0075 |
| Age | 0.95 | 0.93 to 0.98 | P < 0.0001 |
| Day of embryo transfer (≥ Day 5 vs < Day 5) | 1.25 | 1.11 to 1.41 | P = 0.0003 |
| Study site[c] | NA | NA | P < 0.0001 |
| **Live birth** | | | |
| Treatment (oral DYD vs MVP) | 1.28 | 1.04 to 1.57 | P = 0.0214 |
| Age | 0.96 | 0.94 to 0.99 | P = 0.0032 |
| Day of embryo transfer (≥ Day 5 vs < Day 5) | 1.27 | 1.12 to 1.44 | P = 0.0002 |
| Study site[c] | NA | NA | P < 0.0001 |

**Fig 3. Meta-analysis of IPD: Influence of significant predictor variables (including treatment) on ongoing pregnancy and live birth (FAS).** CI, confidence interval; DYD, dydrogesterone; FAS, full analysis sample; IPD, individual participant data; MVP, micronized vaginal progesterone; NA, not applicable; OR, odds ratio. [a]Adjusted for age, study site, and day of embryo transfer. [b]At 12 weeks of gestation. [c]75 sites in the dataset, resulting in 74 ORs and 95% CIs.

## Safety–IPD

**Maternal adverse events.** In the overall population of the two studies with suitable IPD, 342 women who received oral dydrogesterone delivered 418 newborns, and 302 women who received MVP delivered 348 newborns (Table 2). The majority of deliveries were at term (≥37 weeks of gestation) in both the oral dydrogesterone and MVP groups (77.8% [266/342] versus 81.8 [247/302], respectively [Table 2]). Overall, the incidence of maternal AEs was similar between the two treatment groups; maternal AEs that occurred in ≥ 2% of subjects from the safety samples are shown in S4 Table in S1 File. The most frequent maternal AEs reported by subjects treated with oral dydrogesterone versus MVP included vaginal hemorrhage (11.6% [120/1,036] versus 9.5% [97/1,023]), miscarriage (not induced abortion) (8.4% [87/1,036] versus 10.3% [105/1,023]), abdominal pain (7.0% [73/1,036] versus 7.7% [79/1,023]), nausea (5.8% [60/1,036)] versus 4.1% [42/1,023]), procedural pain (5.4% [56/1,036] versus 5.7% [58/1,023]), migraine/headache (4.5% [47/1,036] versus 4.9% [50/1,023]), and vomiting (4.3% [45/1,036] versus 3.7% [38/1,023]).

**Obstetric outcome and fetal/newborn adverse events.** Overall, there were 418 newborns whose mother received oral dydrogesterone, and 348 newborns whose mother received MVP (Table 2).

The impact of treatment group on multiple birth rates was also assessed. In the oral dydrogesterone group, 78.1% (267/342) of mothers had singletons, 21.6% (74/342) had twins, and

**A**

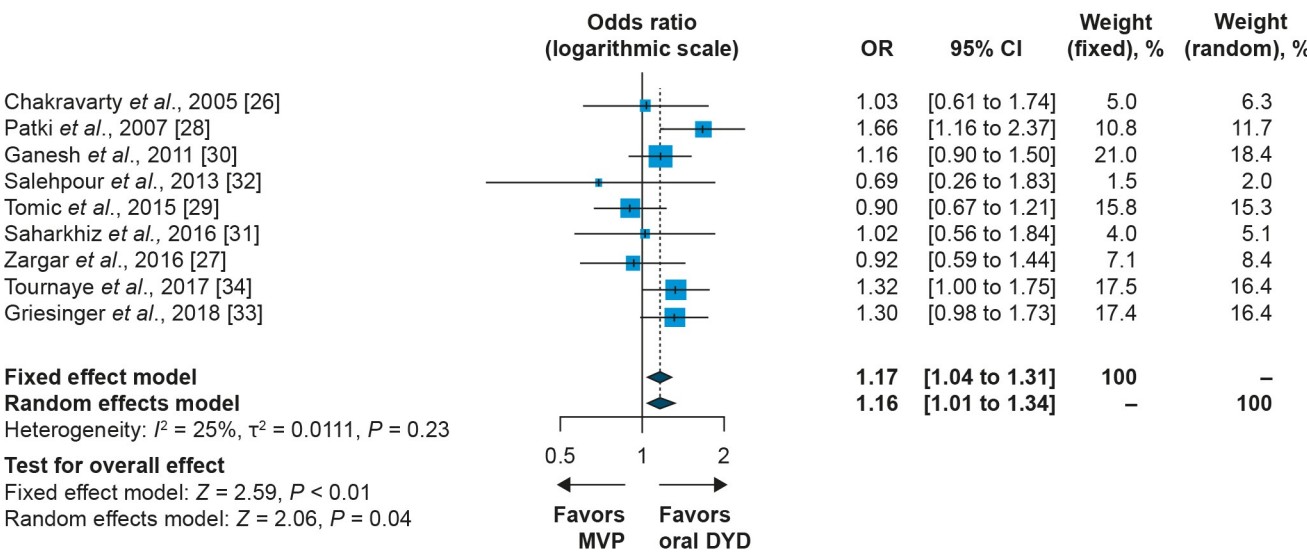

**B**

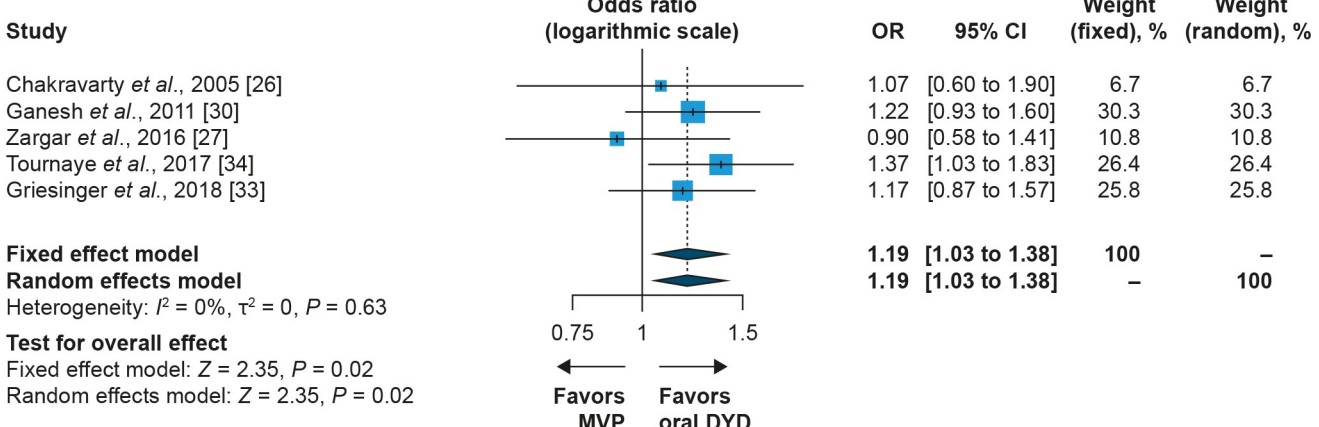

**Fig 4.** Fixed effect and random effects model meta-analysis of IPD and aggregate data: odds ratio for (A) pregnancy rate and (B) live birth rate (oral dydrogesterone versus MVP). CI, confidence interval; DYD, dydrogesterone; MVP, micronized vaginal progesterone; OR, odds ratio.

0.3% (1/342) had triplets (Table 2). In the MVP group, 85.1% (257/302) of mothers had singletons, 14.6% (44/302) had twins, and 0.3% (1/302) had triplets (Table 2).

Overall, 20.3% (131/644) of deliveries were preterm (more than 22 and less than 37 weeks of gestation, based on the date of embryo transfer), with 58.0% (76/131) occurring in the oral dydrogesterone group and 42.0% (55/131) in the MVP group (Table 2). These 131 preterm deliveries resulted in 205 premature newborns, with 58.0% (119/205) and 42.0% (86/205) occurring in the oral dydrogesterone and MVP groups, respectively (S5 Table in S1 File). Among the preterm deliveries, 45.0% (59/131) occurred with singleton pregnancies and 55.0% (72/131) with multiple pregnancies (Table 2). For singletons, the birth weight (mean ± SD) was 3.2 ± 0.6 kg and 3.2 ± 0.5 kg, whereas for multiple births, the birth weight (mean ± SD) was 2.4 ± 0.5 kg and 2.3 ± 0.6 kg in the oral dydrogesterone and MVP groups, respectively (S5

Table in S1 File). Overall, there were 179 newborns with low birth weight (< 2500g), with 26.6% (111/418) and 19.5% (68/348) of all newborns occurring in the oral dydrogesterone and MVP groups, respectively (S5 Table in S1 File). Not surprisingly, the majority of newborns with low birth weight were detected among preterm multiples (twins/triplets) (58.1% [104/179]); 70.2% (78/111) and 82.4% (56/68) of newborns with low birth weight were preterm in the oral dydrogesterone and MVP groups, respectively (S5 Table in S1 File). Among the newborns born at term and with low birth weight, in the dydrogesterone group (29.7% [33/111]) they were predominantly multiples (twins/triplets) compared to singletons (23.4% [26/111] versus 6.3% [7/111], respectively) (S5 Table in S1 File). For MVP, there were 17.6% (12/68) of newborns with low birth weight born at term, and they were more evenly distributed between multiples (twins/triplets) and singletons (10.3% [7/68] versus 7.4% [5/68], respectively) (S5 Table in S1 File).

Overall, the incidence of AEs between the oral dydrogesterone and MVP groups in the newborn populations was similar (S5 Table in S1 File). The most frequent newborn AEs in the oral dydrogesterone versus MVP group included neonatal jaundice (2.9% [12/418] versus 2.9% [10/348]), and neonatal respiratory distress syndrome (2.4% [10/418] versus 3.2% [11/348]), respectively (S5 Table in S1 File). In the two studies with suitable IPD, congenital, familial, and genetic disorders were identified in 35 pregnancies comprising 24 newborns and 11 selective terminations due to malformations as shown in S6 Table in S1 File. Overall, the incidence of congenital, familial, and genetic disorders was similar with 19 cases in the oral dydrogesterone group and 16 cases in the MVP group. Congenital heart disease occurred in 8 and 11 cases respectively, as multiple cardiac-related AEs could occur in a single newborn/fetus.

## Discussion

In order to aggregate the available evidence on the efficacy and safety of oral dydrogesterone and MVP for luteal phase support in fresh-cycle IVF, and to identify prognostic factors of positive clinical outcomes (pregnancy rate and live birth rate), a one-step meta-analysis of IPD was conducted using a combined dataset from RCTs. This analysis identified that treatment was an independent significant prognostic factor for ongoing pregnancy and live birth, whereby oral dydrogesterone was associated with a significantly higher ongoing pregnancy rate and live birth rate than MVP administered as capsules or as a gel.

The meta-analysis of IPD identified other prognostic factors, besides luteal phase support treatment, that influenced the likelihood of ongoing pregnancy and live birth; these included maternal age, day of embryo transfer, and study site. The meta-analysis of IPD also tested for interactions between luteal phase support treatment and the other prognostic factors identified, but no statistically significant interactions were identified. Thus, the associations of maternal age, day of embryo transfer and study site with clinical outcome were not significantly different between the treatment groups.

In this study, there were no differences in the number of embryos transferred, BMI, or the use of ICSI between the two groups, and these were not found to be independent prognostic factors that influenced the likelihood of ongoing pregnancy or live birth.

Other factors that may influence pregnancy outcomes, but were not systematically collected in Lotus I and II, are ovarian reserve [42], ovarian stimulation regimens, gonadotropin-releasing hormone agonist or antagonist usage, oocyte maturation trigger, and IVF clinical embryology data [43]. Based on the non-availability of such study data, any possible interactions between these factors and luteal phase support remain elusive. However, since the Lotus program enrolled over 2000 patients in 14 countries and was shown to be adequately randomized,

it may be assumed that it reflects the general clinical practice of fresh-cycle IVF, where both agonist and antagonist cycles were represented.

The safety outcomes included in this meta-analysis of IPD (pooled results analyzed using descriptive statistics) were similar between the oral dydrogesterone and MVP groups in both the maternal and fetus/newborn populations. Specific complications, such as low birth weight and preterm birth, are known to be associated with assisted reproductive technology. In a 2017 review by *Kushnir, et al.* [44], low birth rate was reported in 12.7% of infants (6.7% for singletons following single embryo transfer and 7.3% following double-embryo transfer, 56.3% for twins, and 97.4% for higher-order multiples), and the preterm delivery rate was reported as 16.6% (10.5% for singletons, 67.3% for twins, and 92.3% for higher-order multiples). The practice of multiple embryo transfer remains prevalent in many countries, driven by a focus on maximizing pregnancy rates since early IVF studies showed that single embryo transfer resulted in lower pregnancy rates per transfer compared to double embryo transfer [45, 46]. However, and importantly, the success rates from single embryo transfer has improved drastically in recent years [45, 47]. Due to the increased risk of maternal and fetal complications associated with multiple gestation from double embryo transfer, there has been an increasing worldwide trend in performing single embryo transfer in patients with good prognosis and utilizing surplus embryos in later frozen-thawed transfers [44].

Corresponding to the overall safety outcomes, in the subgroup analysis of congenital heart disease among newborns and selective terminations due to malformations, similar findings between the oral dydrogesterone and MVP groups in both the maternal and fetus/newborn populations were noted. This is in sharp contrast to a retrospective, case-controlled study from the Gaza region [48] that reported a possible risk for newborn malformations after *in utero* dydrogesterone exposure. Notably, the study by *Zaqout et al.* [48] did not adhere to scientific principles of epidemiological research, such as study base, deconfounding, and comparable accuracy [49]; as a result, no causal relationships should be established from the study as detailed in a recent review article [18]. On the contrary, the available evidence indicates that oral dydrogesterone has a well-established safety profile [24, 50] from long-standing clinical use since the early 1960s, and no new safety concerns were identified for the mother or the fetus in this large IPD set from the two Phase III, randomized controlled trials reported herein.

To date, no other meta-analyses of IPD have been conducted comparing oral dydrogesterone with MVP. Although previous meta-analyses of aggregate data have been conducted [5, 51, 52], none have included both the large Phase III Lotus I and II clinical studies. A Cochrane systematic review and meta-analysis identified a potential positive treatment effect of oral dydrogesterone versus micronized progesterone on clinical pregnancy rates in small studies using fresh-cycle IVF protocols [5]. Another meta-analysis identified a trend for a positive treatment effect of oral dydrogesterone versus MVP in fresh IVF cycles, although it was not statistically significant [51]. Most recently, a meta-analysis was conducted in studies that used fresh or artificial frozen-thawed IVF protocols [52]; however, this study did not take into account the clinical heterogeneity that may exist due to the key endocrinological differences between fresh and artificial frozen-thawed IVF protocols (such as the presence or absence of a corpus luteum) [53], and therefore, the results need to be interpreted with caution.

A recent systematic review and meta-analysis concluded that intramuscular progestogen administration for luteal phase support in fresh-cycle IVF provided greater clinical benefit versus other routes of administration [54]. In that meta-analysis, it was reported that the mean pregnancy rates increased from 14.7% for untreated women to 30.7%, 36.4%, 36.6%, and 44.0% after oral, vaginal, subcutaneous, and intramuscular progestogen supplementation, respectively [54]. Not surprisingly, the optimal time to achieve higher clinical pregnancy rates is to commence luteal phase support between oocyte retrieval and embryo transfer (OR, 1.31),

with oocyte retrieval +1 day reported as being most beneficial. Similarly, clinical pregnancy rates were found to be equivalent when progestogen supplementation was given during a period between ≤ 3 weeks and up to 12 weeks (OR, 1.06) of gestation [54]. This is in line with findings from eight of the nine studies included in the present meta-analysis. It is important to note that a binomial logistic regression model was used in the recent meta-analysis, which does not allow for comparisons of treated versus control groups and also does not allow for comparisons of heterogeneity between studies; additionally, both retrospective and non-randomized trials were included [54]. Since it is known that pregnancy rates from IVF are very different between Europe and North America [55], considerable heterogeneity probably existed between the studies included in the recent meta-analysis [54], which was not taken into account. Importantly, the key Lotus II study was not included in this meta-analysis [54]; furthermore, ongoing pregnancy rates from the meta-analysis of IPD after oral dydrogesterone supplementation presented herein (38.1%) far exceed those reported in the above mentioned meta-analysis [54].

Meta-analyses of IPD have many potential advantages compared with meta-analyses of aggregate data. Importantly, IPD facilitates standardization of analyses across studies, meaning that the findings are more reliable than aggregate data [35]. It also increases the precision of treatment effects by adjusting for important prognostic baseline factors [35]. Notably, in this meta-analysis of IPD, oral dydrogesterone had a significant treatment effect versus MVP for ongoing pregnancy rate (OR, 1.32; 95% CI, 1.08 to 1.61), while the meta-analysis of aggregate data using the same two Lotus studies did not show a significant treatment effect (OR, 1.19; 95% CI, 0.99 to 1.44) although the trend was similar. Furthermore, aggregate data may be presented differently across studies, and this may increase the risk of publication bias and selective reporting [35]. Finally, the effect of missing data at the patient level can affect pooled estimates in aggregate data meta-analyses [56].

Despite the potential advantages, there are limitations to using IPD (such as the lack of available or suitable IPD from the studies of interest) [35]. In this study, despite nine studies being eligible for inclusion, only two had suitable IPD available. Therefore, a meta-analysis combining IPD results from the Tournaye *et al.* [34] and Griesinger *et al.* [33] studies with the aggregate data from the other seven studies using the two-stage approach was conducted. Importantly, the results from this meta-analysis also showed statistically significant differences for both pregnancy rate and live birth rate in favor of oral dydrogesterone versus MVP.

Finally, a meta-analysis of aggregate data for all nine studies was also conducted, which identified that dydrogesterone showed numerically higher pregnancy rates and live birth rates versus MVP, although the results were not statistically significant. For both types of meta-analyses that included aggregate data conducted in this study, as well as previously reported meta-analyses [5, 51, 52], a limitation was that many of the included studies had an unclear methodology, reported pregnancy rates at different timepoints, and used different doses of oral dydrogesterone and MVP. While the meta-analysis of IPD only included two studies, they were large Phase III clinical trials, with robust methodology, and consistent dosing of oral dydrogesterone [33, 34]. Therefore, the meta-analysis of IPD may be considered the most robust currently available estimate of the underlying efficacy differences between MVP and oral dydrogesterone.

The rationale for choosing the oral route over the vaginal route is based on patient and physician preference. Aside from the differences in efficacy between oral dydrogesterone and MVP identified in this study, the administration route of oral dydrogesterone may be advantageous as patients usually prefer to use oral preparations compared to vaginal ones [26, 57, 58]. This may be due to the overall inconvenience [59], administration-related side effects [14, 15], and cultural barriers associated with using MVP [9]. Although MVP may be the most prescribed route of luteal phase support administration [7], it may not be the most preferred

route of administration since women often find vaginal preparations less appealing than oral administration, particularly due to comfort issues [60]. Importantly, improvement in interdisciplinary scientific efforts may be required to increase our understanding of patient needs and preferences [61].

Earlier, we put forward a hypothesis that MVP concentrations in the vagina may alter the local microbiota, which has become a recent focus of interest in the context of IVF [23, 62]. To address this question, some of the authors herein are presently conducting a prospective, randomized, double-blind, double-dummy, two-arm, cross-over study in healthy oocyte donor volunteers to assess if there are differences in vaginal microbiota between groups taking oral dydrogesterone or MVP [63].

It is also important to note that the dosing frequency of oral dydrogesterone (TID) was the same as that used for MVP in four out of the nine studies in this review [26, 28, 30, 34]. As an alternative to MVP, dydrogesterone has high oral bioavailability and selectivity for progesterone receptors [18, 25]; this allows for effective oral administration and circumvents the side effects related to intravaginal administration (such as irritation and discharge) [14, 15]. Another potential advantage of oral dydrogesterone is its cost effectiveness, which is higher than MVP capsules in China and Russia, as shown by a lower cost per live birth in these countries [23, 64, 65].

Luteal phase support regimens in IVF have, to a large degree, evolved empirically. As such, sufficiently powered RCTs of high methodological quality, which are able to detect clinically relevant outcome differences between drugs, dosages, and routes of administration with sufficient confidence have mostly been lacking [5]. It is noteworthy that the largest luteal phase support trial program conducted so far [33, 34], designed as a non-inferiority trial program, now indicates that the standard of care, MVP, may be associated with suboptimal IVF outcomes. The underlying reason for vaginal progesterone being suboptimal when compared with an orally active progestogen is not yet understood; however, it may be speculated that it could be due to an insufficient systemic exposure of the patients to the progestogenic compound and/or the specific mode of action of dydrogesterone, or specific features of dydrogesterone [18].

Our analysis identified that oral dydrogesterone was associated with a significantly higher ongoing pregnancy rate and live birth rate than MVP administered as capsules or as a gel. The meta-analysis of IPD indicated that per 1000 women treated with oral dydrogesterone versus MVP, 381 and 314 achieved an ongoing pregnancy, respectively. This finding is postulated to have implications for the standard of care in IVF treatment. Given the widespread use of MVP and the large number of IVF/ICSI cycles performed worldwide, further research into existing progestogens and administration routes for luteal phase support is warranted, where patient acceptability is taken into consideration in addition to efficacy and safety.

## Supporting information

**S1 File.**
(DOCX)

## Acknowledgments

We would like to thank all the principal investigators from each of the Lotus program study sites and the Lotus I and/or II co-authors Gennady T. Sukhikh (Russia), Ameet Patki (India), Bharati Dhorepatil (India), Dong-Zi Yang (China), and Zi-Jiang Chen (China) for their valuable support.

## Author Contributions

**Conceptualization:** Georg Griesinger, Christophe Blockeel, Elke Kahler, Claire Pexman-Fieth, Jan I. Olofsson, Stefan Driessen, Herman Tournaye.

**Data curation:** Georg Griesinger, Elke Kahler, Claire Pexman-Fieth, Jan I. Olofsson, Stefan Driessen.

**Formal analysis:** Georg Griesinger, Elke Kahler, Claire Pexman-Fieth, Jan I. Olofsson, Stefan Driessen.

**Funding acquisition:** Claire Pexman-Fieth.

**Investigation:** Georg Griesinger, Christophe Blockeel, Elke Kahler, Claire Pexman-Fieth, Jan I. Olofsson, Stefan Driessen, Herman Tournaye.

**Methodology:** Georg Griesinger, Christophe Blockeel, Elke Kahler, Claire Pexman-Fieth, Jan I. Olofsson, Stefan Driessen, Herman Tournaye.

**Project administration:** Georg Griesinger, Claire Pexman-Fieth, Jan I. Olofsson.

**Resources:** Georg Griesinger, Claire Pexman-Fieth, Jan I. Olofsson.

**Software:** Elke Kahler, Stefan Driessen.

**Supervision:** Georg Griesinger, Claire Pexman-Fieth.

**Validation:** Georg Griesinger, Christophe Blockeel, Elke Kahler, Claire Pexman-Fieth, Jan I. Olofsson, Stefan Driessen, Herman Tournaye.

**Visualization:** Georg Griesinger, Christophe Blockeel, Elke Kahler, Claire Pexman-Fieth, Jan I. Olofsson, Stefan Driessen, Herman Tournaye.

**Writing – original draft:** Georg Griesinger, Claire Pexman-Fieth.

**Writing – review & editing:** Georg Griesinger, Christophe Blockeel, Elke Kahler, Claire Pexman-Fieth, Jan I. Olofsson, Stefan Driessen, Herman Tournaye.

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
