## [Decision Letter · Decision Letter 0]

11 Sep 2020

PONE-D-20-24278

Dydrogesterone as an oral alternative to vaginal progesterone for IVF luteal phase support: A systematic review and individual participant data meta-analysis

PLOS ONE

Dear Dr. Griesinger,

Thank you for submitting your manuscript to PLOS ONE. After careful consideration, we feel that it has merit but does not fully meet PLOS ONE’s publication criteria as it currently stands. Therefore, we invite you to submit a revised version of the manuscript that addresses the points raised during the review process.

The present review with meta-analysis reports interesting information on an important aspect of assisted reproduction techniques. From a methodological point of view, the study appears to be conducted appropriately to support the conclusions. Although one or more Authors are affiliated to Abbott, data and conclusions appear to be presented objectively and in an unbiased manner.

According to the AMSTAR guidelines, the quality of the systematic review is appropriate. However, both reviewers raised some interesting questions that the authors are encouraged to consider.

In addition, I ask on the basis of which criteria the studies have been sorted in the forest plots (it is generally suggested to order the studies in chronological order or by effect size).

We look forward to receiving your revised manuscript.

Kind regards,

Alessio Paffoni, PhD

Academic Editor

PLOS ONE

Journal Requirements:

2. We note that the flowchart you included (Figure 1) does not include the reasons you excluded studies from analysis. Please revise the figure to include the reasons for exclusion. Please also revise the figure to include how many studies were excluded for each reason.

"G.G.'s institution has received investigator fees from Abbott during the conduct of the

Lotus I and Lotus II studies. Outside of this submitted work, G.G. has received nonfinancial

support from MSD, Ferring, Merck Serono, IBSA, Finox, TEVA, Glycotope and

Gedeon Richter, as well as personal fees from MSD, Ferring, Merck Serono, IBSA,

Finox, TEVA, Glycotope, Vitrolife, NMC Healthcare, ReprodWissen, Biosilu, Gedeon

Richter and ZIVA.

C.B.’s institution has received investigator fees from Abbott during the conduct of the

Lotus I and Lotus II studies. C.B. is the President of the Belgian Society of

Reproductive Medicine (unpaid) and Section Editor of Reproductive BioMedicine

Online. C.B. has received grants from Ferring, participated in an MSD-sponsored trial

and has received consultancy fees from Ferring, MSD, BioMérieux, Abbott and Merck.

E.K. is an employee of Abbott Laboratories GmbH, Hannover, Germany and owns

shares of Abbott.

C.P.-F. is an employee of Abbott GmbH & Co. KG, Wiesbaden, Germany and owns

shares in Abbott.

J.I.O. is an employee of Abbott Products Operations AG, Allschwil, Switzerland.

S.D. is an employee of Abbott Healthcare Products BV, Weesp, The Netherlands and

owns shares in Abbott.

H.T.'s institution has received investigator fees from Abbott during the conduct of the

Lotus I and Lotus II studies. H.T.’s institution has received grants from Merck, MSD,

Goodlife, Cook, Roche, CooperSurgical, Besins, Ferring and Allergan. H.T. has

received consultancy fees from Gedeon Richter, Merck, Ferring, Abbott and ObsEva."

Reviewers' comments:

Reviewer's Responses to Questions

**Comments to the Author**

1. Is the manuscript technically sound, and do the data support the conclusions?

Reviewer #1: Yes

Reviewer #2: Yes

2. Has the statistical analysis been performed appropriately and rigorously? 

Reviewer #1: I Don't Know

Reviewer #2: Yes

3. Have the authors made all data underlying the findings in their manuscript fully available?

Reviewer #1: Yes

Reviewer #2: Yes

4. Is the manuscript presented in an intelligible fashion and written in standard English?

Reviewer #1: Yes

Reviewer #2: Yes

5. Review Comments to the Author

Reviewer #1: This is a well written manuscript that presents interesting results related to one of the most controversial and fascinating aspects of IVF treatments: the support of the luteal phase. The work describes a method of meta-analysis (IPD), which could be not so familiar to several readers. However, the strengths and limitations of the method are well described. The results may have important clinical implications. The review is available in the attachment.

Reviewer #2: Thank you for the opportunity to review this meta-analysis examining oral vs vaginal progesterone in fresh IVF cycles. Overall, this paper is well written with good information. The study, however, may be biased, given the contribution of the pharmaceutical company. I have several concerns/comments that should be addressed:

1. The gold standard for progesterone luteal phase supplementation is PIO (progesterone in oil). In the introduction, the authors should state why they chose to study vaginal administration vs oral instead of oral vs PIO

2. All figures/tables should be at the end of the manuscript, instead of being inserted in the middle

3. There are several different doses of oral progesterone used in the various studies. Why was this done and did this change any of the findings?

6. PLOS authors have the option to publish the peer review history of their article (what does this mean?). If published, this will include your full peer review and any attached files.

Reviewer #1: No

Reviewer #2: No

---

## [Author Response · Author response to Decision Letter 0]

25 Sep 2020

Due to the formatting restrictions of this plain text box, and the use of colour-coding and strikethroughs within the Response to Reviewers document, it is not possible to copy and paste the information here whilst also retaining the original formatting, which aids the explanation of each response. 

Please refer to the 'Griesinger et al_Response to Reviewers' document uploaded with this resubmission.

---

## [Decision Letter · Decision Letter 1]

8 Oct 2020

Dydrogesterone as an oral alternative to vaginal progesterone for IVF luteal phase support: A systematic review and individual participant data meta-analysis

PONE-D-20-24278R1

Dear Dr. Griesinger,

We’re pleased to inform you that your manuscript has been judged scientifically suitable for publication and will be formally accepted for publication once it meets all outstanding technical requirements.

Kind regards,

Alessio Paffoni, PhD

Academic Editor

PLOS ONE

Additional Editor Comments (optional):

Reviewers' comments:

Reviewer's Responses to Questions

**Comments to the Author**

1. If the authors have adequately addressed your comments raised in a previous round of review and you feel that this manuscript is now acceptable for publication, you may indicate that here to bypass the “Comments to the Author” section, enter your conflict of interest statement in the “Confidential to Editor” section, and submit your "Accept" recommendation.

Reviewer #1: All comments have been addressed

Reviewer #2: All comments have been addressed

2. Is the manuscript technically sound, and do the data support the conclusions?

Reviewer #1: Yes

Reviewer #2: Yes

3. Has the statistical analysis been performed appropriately and rigorously? 

Reviewer #1: I Don't Know

Reviewer #2: Yes

4. Have the authors made all data underlying the findings in their manuscript fully available?

Reviewer #1: Yes

Reviewer #2: Yes

5. Is the manuscript presented in an intelligible fashion and written in standard English?

Reviewer #1: Yes

Reviewer #2: Yes

6. Review Comments to the Author

Reviewer #1: The authors improved the quality of their work and clarified the doubts raised by this reviewer. Clinicians and researchers will read the manuscript with interest.

Reviewer #2: The authors have sufficiently answered the questions and concerns posed. The manuscript is now in an acceptable form.

7. PLOS authors have the option to publish the peer review history of their article (what does this mean?). If published, this will include your full peer review and any attached files.

Reviewer #1: No

Reviewer #2: No

---

## [Editor Report · Acceptance letter]

14 Oct 2020

PONE-D-20-24278R1 

Dydrogesterone as an oral alternative to vaginal progesterone for IVF luteal phase support: A systematic review and individual participant data meta-analysis 

Dear Dr. Griesinger:

I'm pleased to inform you that your manuscript has been deemed suitable for publication in PLOS ONE. Congratulations! Your manuscript is now with our production department. 

Kind regards, 

on behalf of

Dr. Alessio Paffoni 

Academic Editor

PLOS ONE